# In Situ Characterization of 17-4PH Stainless Steel by Small-Angle Neutron Scattering

**DOI:** 10.3390/ma16165583

**Published:** 2023-08-11

**Authors:** Shibo Yan, Zijun Wang, Tianfu Li, Zhong Chen, Xiaoming Du, Yuntao Liu, Dongfeng Chen, Kai Sun, Rongdeng Liu, Bing Bai, Xinfu He, Kaitai Liu, Shuanzhu Wang

**Affiliations:** 1China Institute of Atomic Energy, Beijing 102413, China; 2School of Materials Science and Engineering, Shenyang Ligong University, Shenyang 110158, China

**Keywords:** 17-4PH, martensitic stainless steel, neutron scattering, in situ SANS, precipitation

## Abstract

17-4PH martensitic steel is usually used as valve stems in nuclear power plants and it suffers from thermal aging embrittlement due to long-time service in a high-temperature and high-pressure environment. Here, we characterized the evolution of microstructures at the nano-scale in 17-4PH steel by in situ small-angle neutron scattering (SANS) with a thermo-mechanically coupled loading device. The device could set different temperatures and tensile so that an in situ SANS experiment could dynamically characterize the process of nanoscale structural changes. The results showed that with increasing thermal aging time, the ε-Cu phase precipitates and grows as the temperature is 475 °C and 590 °C, and the ε-Cu phase is spherical at 475 °C but became elongated cylinders at 590 °C. Moreover, the loading stress could aid in the growth of the ε-Cu phase at 475 °C.

## 1. Introduction

17-4 stainless steel is a type of martensitic precipitation-hardening stainless steel, that was originally developed by Armco in the 1940s. Due to its excellent wear resistance, high-temperature resistance, and high toughness, 17-4PH martensitic steel has been widely used as a structural material in nuclear power plants, such as for valve stems. Nevertheless, after prolonged service at temperatures of approximately 300 °C, the stem of the 17-4PH martensitic steel valve is subject to thermal aging and embrittlement, which can lead to valve fractures and compromise the stable operation of nuclear power plants [1]. Therefore, examining the thermal aging effect of martensitic stainless steel is of paramount importance to ensure the safe operation of nuclear power plants.

Numerous investigations have been conducted to explore the transformations in the microstructure and mechanical properties of 17-4PH stainless steel following solution and aging treatments [2,3,4,5]. In terms of microstructure, spinodal decomposition occurs on the martensite matrix after long-term service. Some long-term and medium-temperature aging experiments show that the G phase, as an intermetallic compound, is usually precipitated with spinodal decomposition [6]. Additionally, nanophase precipitation occurs during the thermal aging process of 17-4PH martensitic steel, with the precipitation and growth of Cu-rich phases acknowledged as a key factor contributing to 17-4PH heat aging embrittlement [7,8]. Aghaie-Khafri et al. studied the phase separation and precipitation process of 17-4PH under long-term aging conditions and found that the precipitation and growth of Cu could be the cause of aging embrittlement. Therefore, the optimum mechanical properties can be obtained by aging at 400 °C for a few hours, mainly due to the precipitation of the coherent nanosized copper precipitates [9]. Yeli et al. investigated the evolution of Cu-rich precipitates from 17-4PH and their effect on material hardening after aging at 480 °C and 590 °C. They found that the Cu-rich precipitates reached a maximum number density and volume fraction within 30 min at 480 °C or 10 min at 590 °C. Then, the existing smaller Cu-rich precipitates coarsened upon prolonged aging, leading to a decrease in number density. Additionally the peak number density at 480 °C was much higher and the growth is much slower than that at 590 °C [10]. It is known that Cu-rich precipitates are formed by a BCC/9R/FCC phase transformation in ferritic or martensitic steels [11,12,13]. The previously coherent spherical Cu-rich precipitates were found to coarsen in size, become cylindrical in shape and segregate along grain boundaries or dislocations after prolonged aging. These nanoscale microstructures substantially affected the material properties.

Currently, researchers often use scanning electron microscopy (SEM), transmission electron microscopy (TEM), and atomic probe methods to study the microstructure of 17-4PH stainless steel [14,15]. However, these approaches are limited to ex situ experiments and the dynamic in situ characterization of samples, which are exceptionally difficult to perform with these techniques. This limitation is significant, as the internal nanoscale microstructure of 17-4PH stainless steel undergoes rapid changes during the initial stages of thermal aging, necessitating in situ characterization to elucidate the evolution mechanism of its nanostructure.

Small-angle neutron scattering (SANS) is a unique technique that can be used to study the nanostructure of metallic materials. Neutrons provide strong penetrability, and SANS can measure a large sample volume (approximately 10 mm^3^~500 mm^3^); it is easy for sample preparation. Additionally, SANS is suitable for loading in situ devices due to the large space of the sample position and high penetration ability of neutrons. The in situ SANS has been widely used to study the evolution of nanoprecipitates inside metals such as γ′ precipitates in Ni-base superalloys, the evolution of precipitates in vanadium micro-alloyed steels, Cu-enriched precipitates in Fe-Cu-Ni model alloys [16,17,18,19]. This study aimed to investigate the morphology and size evolution of Cu-enriched precipitates in 17-4PH martensitic steel with different aging temperatures and tension using in situ small-angle neutron scattering. As a result, we could develop a better understanding of the dynamic formation mechanism of the nanoprecipitates in 17-4PH martensitic steel.

## 2. Materials and Methods

### 2.1. Materials

The material used is a 17-4PH precipitation hardening martensitic stainless-steel hot-rolled sheet, and its chemical composition is shown in Table 1. The sheet was solution-treated at 1050 °C for 2 h, quenched with water at room temperatures, and treated by tempering at 600 °C for 4 h before air cooling. The sheet treated by the above heat treatment was processed into a uniaxial creep testing sample in tension by GB/T 2039-2012 [20] Chinese national standard (as shown in Figure 1). The samples were categorized into 3 groups according to different temperatures and tensions, and an in situ thermo-mechanically coupled loading device was used to measure by small-angle neutron scattering.

### 2.2. Small-Angle Neutron Scattering

The incident neutron beam is normally generated by the reactor or accelerator in small-angle neutron scattering experiments. After the incident neutron beam bombards the sample, the detector records the ratio of the scattering intensity to the incident intensity; specifically, the ratio is the micro differential cross-section at the solid angle dΩ at the distance r from the sample. The absolute scattering intensity as a micro-differential cross-section per unit volume of sample.
$$I{\left(Q\right)}_{CAL}=\phi AdT_{sample+cell}{\left(\frac{d\Sigma (Q)}{d\Omega }\right)}_{sample}\Delta \Omega \varepsilon t$$
where *I* is the scattered intensity, ϕ is the incident neutron flux, A is the sample area, d is the sample thickness, ΔΩ is the solid angle of each pixel, ε is the detector efficiency, and t is the counting time. The scattering intensity is factorized during data analysis and expressed as a product of scale factors, shape factors, and structural factors.
*I* = *NP*(*Q*)*S*(*Q*)
where N is the scale factor, $A=n_{p}{\left(\rho_{p}-\rho_{s}\right)}^{2}$ and *n_p_* is the particle number density. The scale factor is affected by the density of the number of media, the contrast between particles and the matrix; *P*(*Q*) is a normalized form factor, which depends only on the shape size of the particle; *S*(*Q*) is the structural factor, which is only affected by the spatial distribution of molecules.

In this study, the nanoprecipitates are approximately spherical or elongated cylinders, and their scattering intensities meet the following formulas
$$I(Q_{sphere})=\frac{scale}{V}{\left[3V(∆\rho )\cdot \frac{\sin(Qr)-Qr\cos(Qr)}{{\left(Qr\right)}^{3}}\right]}^{2}+background\;I(Q_{cylinder})=\frac{scale}{V}\int_{0}^{\frac{\pi }{2}}F^{2}(Q,\alpha )\sin\alpha d\alpha +background$$
where F(Q,α) is defined as follows:$$F(Q,\alpha )=2(∆\rho )V\frac{\sin(\frac{1}{2}QL\cos\alpha )}{\frac{1}{2}QL\cos\alpha }\frac{J_{1}(QR\sin\alpha )}{QR\sin\alpha }$$
where α is the angle between the axis of the cylinder, V is the volume of the cylinder or sphere, L is the length of the cylinder, R is the radius of the cylinder, Δρ is the scattering length density difference between the scatterer and the solvent, and J_1_ is the first-order Bessel function. $\alpha \vec{Q}V=\pi R^{2}LLR\Delta \rho J_{1}$ [21].

The in situ SANS measurements were carried out by the small-angle neutron-scattering instrument of the China Advanced Research Reactor [22]. The SANS data were collected at the sample-to-detector distance of 3 m. The speed of the velocity selector was set to 4500 rpm and the corresponding neutron wavelength was 0.60 nm. The in situ thermo-mechanically coupled loading device was utilized for the measurement, and could load up to 20 kN tensile force with a temperature of up to 800 °C, the temperature control accuracy was better than ±1 °C. The sample experiment information is shown in Table 2.

The data preprocessing software Fit2D v18.002 was used to perform background correction and normalization [23], and SasView was used to fit the model of the SANS data [24].

### 2.3. Transmission Electron Microscopy

The corresponding microstructures were examined by using optical microscopy and TEM. The sample is S1 and S3 which after 475 °C and 590 °C thermal aging. Thin foils were prepared for TEM from 0.25 mm thick which slit from the S1 and S3 sample. Then, followed by standard grinding and twin-jet electropolishing procedures, the sample was examined using an FEI-TALOS-F200X TEM (Waltham, MA, USA) operating at 200 kV. 

## 3. Results and Discussion

The data of the in situ SANS measurements from different heat-treated temperature and tension samples are presented in Figure 2. They were generated independently every 15 min under the condition of good statistical quality. Changes in scattering intensity originated from the inhomogeneity of the material at the nanoscale. As shown in Figure 2, the scattered signal of the sample significantly changed every 15 min, indicating that the morphology and size of the nano-precipitated phase inside the material evolved. For 17-4PH martensitic steel, solution treatment in the temperature range of 300~620 °C formed a variety of nano-precipitation phases. For example, ε-Cu, NbC, M_23_C_6_, and ferrite formed the Cr-rich α′ phase and Fe-rich α phase through spinodal decomposition. The nano-precipitations caused age hardening. Depending on the thermal aging time and temperature, the morphology and size of precipitated phases at the nanoscale also varied. After a short aging time below 560 °C, the precipitated phase in the alloy is mainly ε-Cu, a small amount of carbide (NbC and M_23_C_6_). Long-term aging (more than 100 h) resulted in the appearance of a Cr-rich α′ phase due to the spinodal decomposition of ferrite. After aging above 560 °C, the thick ε-Cu, carbide, G phase and σ phase precipitated in the alloy.

As shown in Figure 2, the scattered signal intensity of the sample has different trends at different temperatures. The signal of the sample in the low Q region (Q < 0.2 nm^−1^) originates from the contribution of the relatively large carbides and grain boundaries inside the material that is larger than 30 nm; the signal in the high-Q region (Q < 0.2 nm^−1^) originates from the inhomogeneity of the small nano-precipitated phase. Studies have shown that spinodal decomposition occurs only under long-term thermal aging (>100 h) [25]. Therefore, the contribution of the scattered signal here mainly comes from the nano-precipitated phase. The overall intensity increases with time when the loading temperature is 475 °C and 590 °C, which means that the nano-precipitate content inside the material increases with the aging time, as shown in Figure 2a,b. The scattering signal suddenly decreases in a short period of time when the sample is heated to 650 °C from 475 °C, as shown in Figure 2a. This characteristic is potentially caused a sharp growth of the precipitated phase due to high-temperature aging; high-temperature aging also causes a significant decrease in the number of precipitated phases, resulting in a sudden weakening of the scattering signal in the measurement Q range.

The morphology and size evolution of the precipitates in the S2 and S3 samples were investigated using TEM after the SANS experiment. Figure 3a shows the TEM micro-graphs of the Cu-rich precipitates in the S1 sample after aging at 475 °C for 3 h, and Figure 3b shows the S3 sample after aging for 3 h at 590 °C. Many precipitates were observed in the 17-4PH steel; clearly, the precipitates in S2 were spherical in shape and those in S3 were ellipsoidal or elongated cylinders. The change in precipitate morphology was actually the transition process from the coherent BCC Cu-rich phase (spherical) to the 9R twin structure (spherical) of the FCC structure (elongated cylinder shape), which was a spontaneous energy-change process. Its driving force was the increase in aging temperature.

To quantitatively describe changes in the nanoscale structure, the SANS data needed to be fitted. According to previous reports, the ε-Cu phase of 17-4PH martensitic stainless steel began to precipitate after 480 °C aging, and was mostly spherical [25,26]. Our TEM experiment can also confirm this result. Therefore, the in situ SANS data of the S1 samples aged at 475 °C for 45–60 min, 105–120 min, and 165–180 min were fitted using a polydisperse sphere model with a normal logarithmic distribution. The fitting requires a selection of appropriate scattering length density. According to the chemical composition, the scattering length density of the Cu precipitated phase was 6.55 × 10^−6^/Å^−2^. The chemical composition of the matrix was Fe71Cr17Ni5Cu5Mn1Si1, and the scattering length density is Fe: 8.024 × 10^−6^/Å^−2^, Ni: 9.408 × 10^−6^/Å^−2^, Cr: 3.027 × 10^−6^/Å^−2^, Cu: 6.550 × 10^−6^/Å^−2^, Mn: −3.013 × 10^−6^/Å^−2^, Si: 2.074 × 10^−6^/Å^−2^; thus, the scattering length density of the matrix was 7.001 × 10^−6^/Å^−2^, and the fitting results are provided in Figure 4 and Table 3. The average radius of the nano-phase inside the S1 sample increased with aging time (Figure 4b), and the polydispersity ratio initially increased and then remained unchanged after 2 h (Table 3). During aging, the average radius of the precipitated phase increased from 1.63 nm to 2.71 nm, and the polydisperse ratio increased from 0.38 to 0.41. This result indicated that ε-Cu inside S1 gradually grew with aging at 475 °C, and the size of spherical nanoprecipitates increased to 5.4 nm with aging for 3 h (Figure 4b). The increase in polydispersity at the beginning of aging meant that the ε-Cu phase grew while new Cu nanoparticles were continuously nucleated and precipitated. The polydispersity remained unchanged after 2 h, indicating that the nucleated precipitations in the later stage of aging significantly decreased.

Habibi et al. studied the aging of 17-4PH steel at 480–630 °C, and found that a higher aging temperature correlated to a larger particle size of the Cu-rich phase, and the morphology changed from a nearly spherical granular to a short rod [27]. In the previous work, we also found that the Cu particles precipitated in the early-stage changes from spherical to short rod shape during the process of 12 h aging at 600 °C [19]. Therefore, the log-normal cylindrical distribution model was used to fit the SANS data of S3 samples in the range of 45–60 min and 165–180 min. The scattering length density of the precipitated phase and the matrix were consistent with before, the fitting results are shown in Figure 5, and the resulting structural parameters are listed in Table 4. The fitting results showed that the average radius and length of the elongated cylinder nanophase inside the S3 sample increased with aging time, the average radius increased from 2.83 nm to 3.01 nm, and the length increased from 15.24 nm to 21 nm; the polydispersity ratio of the sample decreased with the aging time, the polydispersity ratio of the average radius decreased from 0.54 to 0.42, and the polydispersity of length decreased from 0.24 to 0.10. The cylindrical model could fit the experimental data well, showing that during the aging process at 590 °C, the Cu phase changed from spherical to cylindrical. With aging time, the size of the precipitates increased, and the polydispersity ratio decreased. During this process, the existing Cu precipitates continued to grow, and smaller precipitates grew faster than larger precipitates. Comparing the results at 475 °C, the growth rate of the precipitation of the Cu phase was faster as the aging temperature increased.

Figure 6 shows the scattering intensity of the S1 sample with aging time. The scattering intensity significantly changed with aging temperature and time. The scattering signal significantly decreased within 15 min when the aging temperature increased to 650 °C; this result indicated that the inhomogeneity of the S1 sample at the nanoscale in the measured Q range was significantly reduced, but the quantity analysis results of the nanophase size could not be obtained by model fitting. Generally, the aging temperature of 17-4PH precipitation hardening stainless steel after solution treatment is 480–620 °C. During this period, a higher aging temperature correlates to a larger size of the ε-Cu precipitated phase. The 40–50 nm ε-Cu phase was produced at 620 °C, and the roughening of the Cu-rich phase led to a decrease in the yield strength and tensile strength of the material [15]. When the temperature increased to 650 °C, the ε-Cu precipitated phase further merged and grew, resulting in a significant decrease in the number of precipitated phases. In addition, 17-4PH martensitic steel aging at 650 °C also caused fine ε-Cu phase solid dissolution back to the matrix, that is the reversion of the alloy led to a significant reduction of ε-Cu phase, both of which sharply reduced the scattering signal in the measurement Q range.

The difference between the S2 sample and the S1 sample is that a uniaxial tensile load of 10 kN is applied to S2 during the aging process, and the calculated tensile stress is 500 MPa according to the sample size. 17-4PH precipitation-hardening martensitic stainless steel has a tensile yield strength of 739 MPa at 500 °C [3]. Therefore, the tensile stress of loading 500 MPa at 475 °C in this experiment did not cause plastic deformation of the sample. Figure 7 shows a comparison of the scattering curves of the S1 and S2 samples. The scattering curves of the S1 and S2 samples did not differ significantly at the early stage of aging, and the scattering curves of the two samples basically overlapped; however, the scattering curves began to differ after loading for 1 h. The Q value was in the range of 0.3 nm^−1^–0.8 nm^−1^, and the scattering intensity of the S2 sample loaded with 10 kN is slightly higher than that of the S1 sample. The scattering data of the S1 and S2 samples at 165–180 min were fitted, the results are shown in Figure 8, and the fitting parameters are listed in Table 5. The average radius of the internal nanophase of the S2 sample was 2.82 nm, which was slightly larger than that of the S1 sample (2.71 nm) at the same load time, while the polydispersity does not change. This result showed that the loading 10 kN tensile stress at 475 °C contributed to the growth of Cu-rich nanophases inside the material. Ratel et al. investigated the mechanism of high-temperature rafting in nickel-based single-crystal superalloys using in situ SANS and showed that the application of strain fields promoted the kinetic process of orientational coarsening in the presence of heat treatment coupled with creep [28,29].

## 4. Conclusions

In situ SANS is a unique and powerful technology that can characterize the evolution process of precipitation in alloys. In this study, in situ SANS and TEM were used to investigate the evolution process of the Cu-rich precipitate in 17-4PH steel with different aging temperatures and tensile forces. The results showed that with increasing aging time, the Cu-rich phase precipitated and grew. In the case of aging for 3 h at 475 °C, the ε-Cu-precipitated phase radius grew from 1.63 nm to 2.71 nm, which could help the growth of the ε-Cu phase when loading 10 kN tensile stress at 475 °C; when aging for 3 h at 590 °C, the ε-Cu-precipitated phase length increased from 15.24 nm to 21 nm. Based on the SANS data analysis, the precipitation growth rate of the Cu phase during the aging process at 590 °C was faster than that at 475 °C. Furthermore, the TEM results showed that the Cu-rich precipitates transformed from spherical to short rods or elongated cylinders, and the change in precipitate morphology was the transition process from the coherent BCC Cu-rich phase to the 9R twin structure to the FCC structure with increasing thermal aging time and temperature. When the aging temperature was increased from 475 °C to 650 °C, the overall scattered intensity suddenly decreased, a relatively large ε-Cu precipitated phase further merged and grew during this time, and small ε-Cu precipitates were dissolved back into the matrix.

## Figures and Tables

**Figure 1 materials-16-05583-f001:**
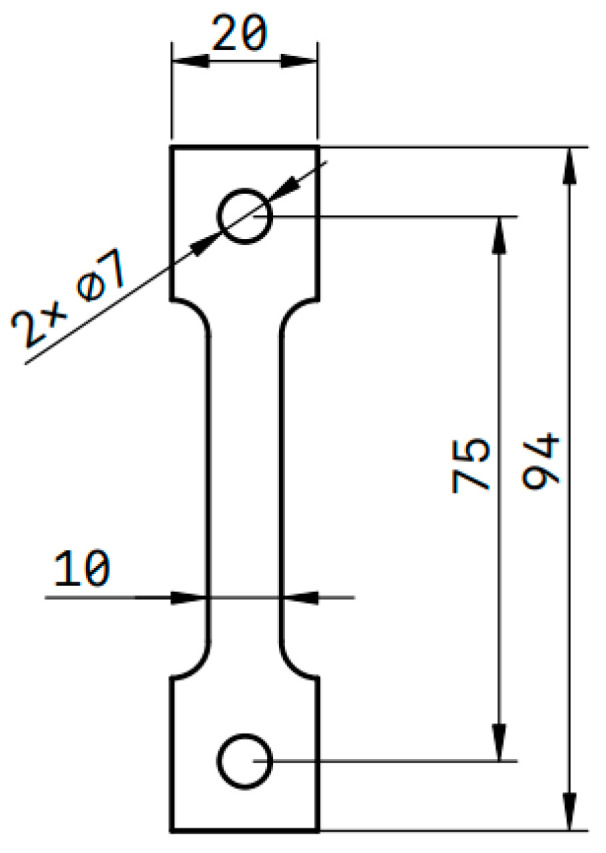
In situ experimental sample size.

**Figure 2 materials-16-05583-f002:**
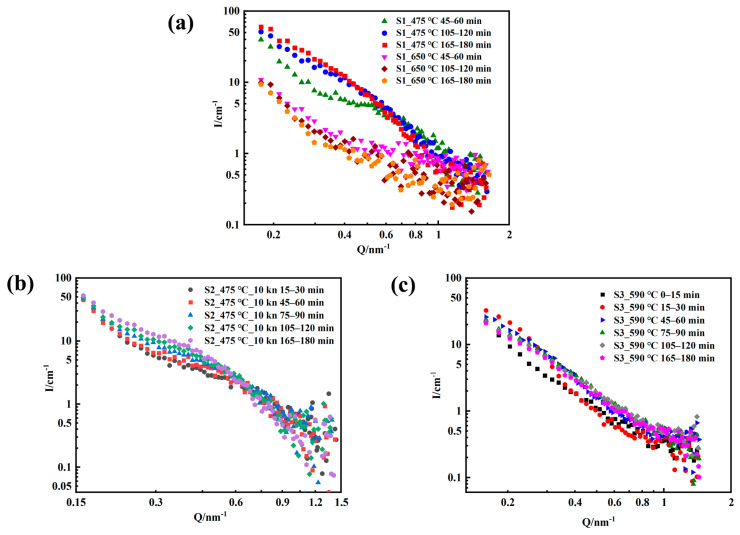
SANS scattering data from S1–S3 (**a**–**c**).

**Figure 3 materials-16-05583-f003:**
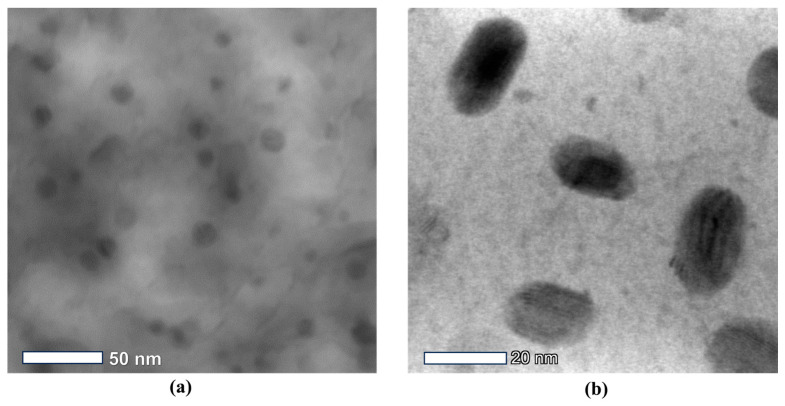
TEM micrographs of the Cu-enriched precipitates in the S2 and S3 samples: (**a**) Bright-field image of the S2 sample after 3 h of aging at 475 °C and (**b**) Bright-field image of the S3 sample after 3 h of aging at 590 °C.

**Figure 4 materials-16-05583-f004:**
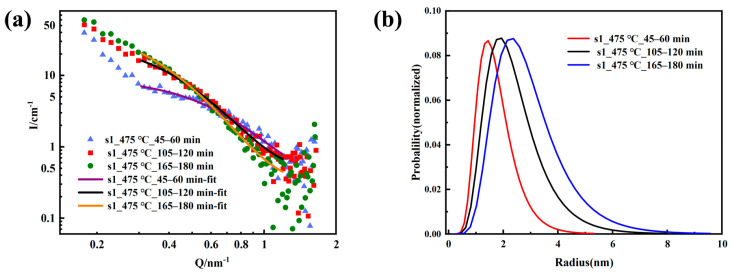
S1 sample fitting curve (**a**) and the size distribution (**b**).

**Figure 5 materials-16-05583-f005:**
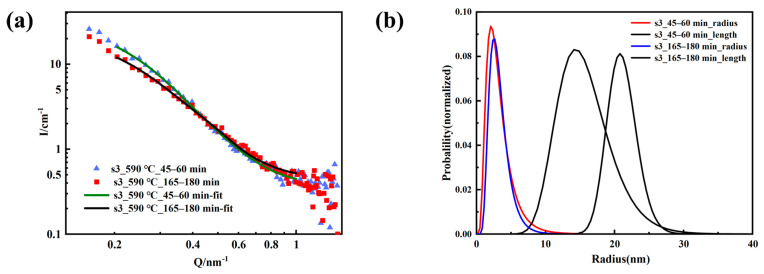
S3 sample fitting curve (**a**) and the size distribution (**b**).

**Figure 6 materials-16-05583-f006:**
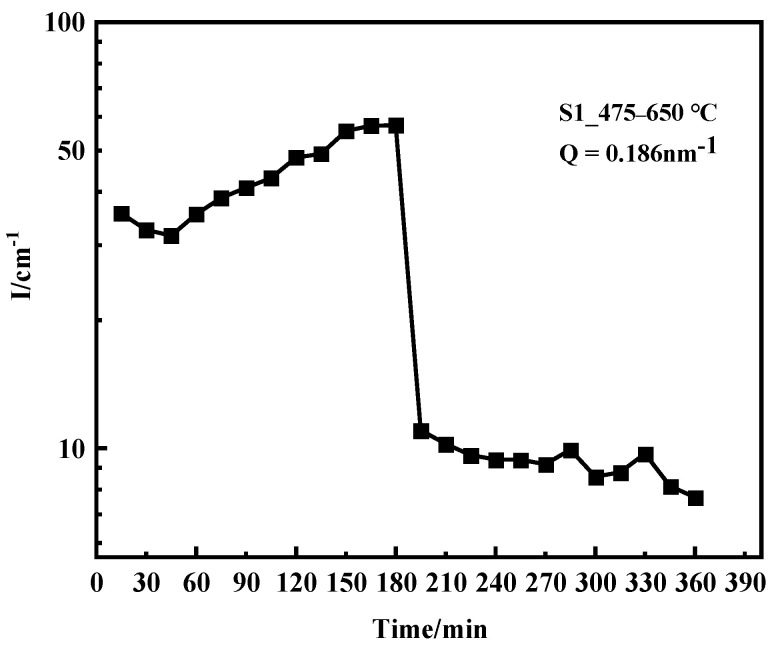
Time-scattering intensity graph of the S1 sample.

**Figure 7 materials-16-05583-f007:**
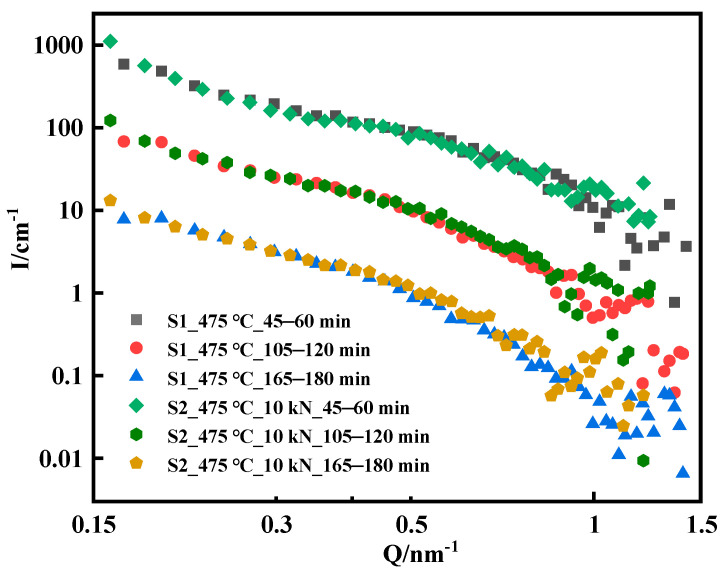
Comparison of the scattering data of S1 and S2. For the data of S1 and S2 at 45–60 min, the scattering intensity is multiplied by 100 and for the data of S1 and S2 at 105–120 min, the scattering intensity is multiplied by 10 such that the difference between S1 and S2 data can be clearly compared.

**Figure 8 materials-16-05583-f008:**
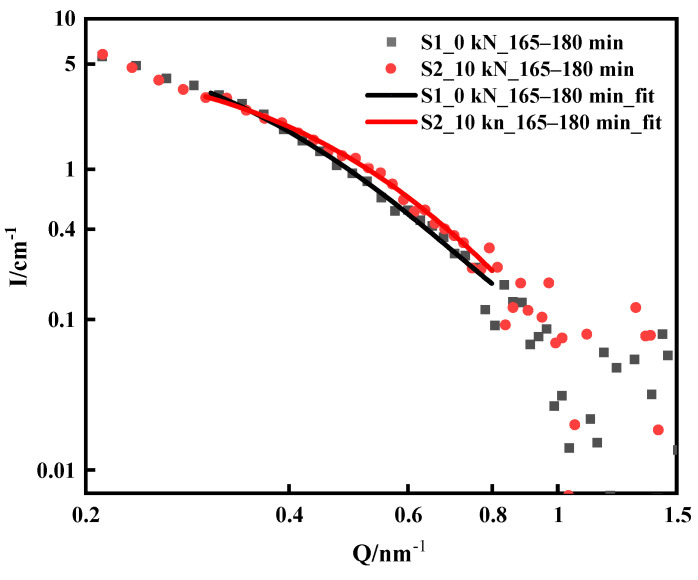
Fitting results at 165–180 min for S1 and S2.

**Table 1 materials-16-05583-t001:** Chemical composition of 17-4PH martensitic stainless steel.

Element	C	Si	Mn	P	S	Cr	Ni	Cu	Nb
Wt/%	<0.07	<1	<1	<0.003	<0.003	15–17	3–5	3–5	0.15–0.45

**Table 2 materials-16-05583-t002:** Experimental information for S1–S3.

No.	Temperature (°C)	Tensile (Kn)	Time (h)
S1	475, 650	0	6 (475 °C—3 h, 650 °C—3 h)
S2	475	10	3
S3	590	0	3

**Table 3 materials-16-05583-t003:** S1 fitting result.

No.	Model	Q Range/nm^−1^	Size/nm	Polydispersity Ratio
S1_45–60 min	Sphere	0.3–1.2	R = 1.63	0.38
S1_105–120 min	Sphere	0.3–1.2	R = 2.22	0.41
S1_165–180 min	Sphere	0.3–1.2	R = 2.71	0.41

**Table 4 materials-16-05583-t004:** S3 fitting result.

No.	Model	Q Range/nm^−1^	Size/nm	Polydispersity Ratio
S3_45–60 min	Cylinder	0.2–1	R = 2.83 L = 15.24	R~0.54 L~0.24
S3_165–180 min	Cylinder	0.2–1	R = 3.01 L = 21.00	R~0.42 L~0.10

**Table 5 materials-16-05583-t005:** S1, S2 fitting result at 165–180 min.

No.	Model	Q Range/nm^−1^	Size/nm	Polydispersity Ratio
S1_165–180 min	Sphere	0.3–0.8	R = 2.71	0.41
S2_165–180 min	Sphere	0.3–0.8	R = 2.82	0.41

## Data Availability

The data that support the findings of this study are available from the corresponding author upon reasonable request.

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
