# Peer review of "In Situ Characterization of 17-4PH Stainless Steel by Small-Angle Neutron Scattering"

_materials, 2023, doi:10.3390/ma16165583_

Round 1

Reviewer 1 Report

The present manuscript is aimed to investigate the morphology and size evolution of Cu-enriched precipitates in 17-4PH martensitic steel with different aging temperatures and tension using in-situ small-angle neutron scattering. Although, the authors have been reported some interesting data, the manuscript must be revised. The detailed problems are following:

1) Although the introduction contains some previous data, but it should be completed.

 2) English texting should be fundamentally revised. There are some grammatical mistakes so that some parts of statements are confusing.

 3) The figures must be improved.

4) In line 225, it has been mentioned that “Figure 6 shows a comparison of the scattering curves of S1 and S2 samples, it can be seen that the scattering curves of S1 and S2 samples did not differ significantly at the early stage of aging, the scattering curves of the two samples basically overlapped, but the scattering curves began to differ after loading for 1 h.”. What is that reason?

 5) The conclusion must be completed. It is not acceptable in present form.

English texting should be fundamentally revised. There are some grammatical mistakes so that some parts of statements are confusing.

Author Response

Thank you for your comments concerning our manuscript entitled “In situ 
characterization of 17-4PH stainless steel by small angle neutron scattering” (materials-2444717). Those comments are all valuable and very helpful for revising and improving our paper, as well as the important guiding significance to our researches. We have studied comments carefully and have made correction which we hope meet with approval. The details of our response are in the PDF.

Reviewer 2 Report

Please check the list of suggestions in the attached file.

Author Response

(The authors gave the same response as above.)

Reviewer 3 Report

The authors use a modern research method (small angle neutron scattering) to study the aging processes in 17-4PH steel.

Considering the limited experience of using this method, it is desirable to compare the obtained results of the study with the results of more proven methods: (scanning electron microscopy, X-ray diffractometry), which also allow monitoring the progress of processes at elevated temperatures.

The authors did not present a hypothesis explaining the transition from spherical copper particles, which have a minimum surface energy, to a short-rod morphology, which has a much larger specific surface area.

The authors talk about the possibility of both the processes of nanophase precipitation during aging at 650°C and the dissolution of already separated phases. It would be interesting to compare these studies with the solubility curves of copper in the ferrite of this steel at elevated temperatures.

Author Response

(The authors gave the same response as above.)

Round 2

Reviewer 1 Report

Accept